# The Cost of Compression: Tight Quadratic Black-Box Attacks on Sketches for $\ell_2$ Norm Estimation

**Sara Ahmadian**
Google Research
sahmadian@google.com

**Edith Cohen**
Google Research and Tel Aviv University
edith@cohenwang.com

**Uri Stemmer**
Tel Aviv University and Google Research
u@uri.co.il

## Abstract

Dimensionality reduction via linear sketching is a powerful and widely used technique, but it is known to be vulnerable to adversarial inputs. We study the *black-box adversarial setting*, where a fixed, hidden sketching matrix $A \in \mathbb{R}^{k \times n}$ maps high-dimensional vectors $\boldsymbol{v} \in \mathbb{R}^n$ to lower-dimensional sketches $A\boldsymbol{v} \in \mathbb{R}^k$, and an adversary can query the system to obtain approximate $\ell_2$-norm estimates that are computed from the sketch.

We present a *universal, nonadaptive attack* that, using $\tilde{O}(k^2)$ queries, either causes a failure in norm estimation or constructs an adversarial input on which the optimal estimator for the query distribution (used by the attack) fails. The attack is completely agnostic to the sketching matrix and to the estimator—it applies to *any* linear sketch and *any* query responder, including those that are randomized, adaptive, or tailored to the query distribution.

Our lower bound construction tightly matches the known upper bounds of $\tilde{\Omega}(k^2)$, achieved by specialized estimators for Johnson–Lindenstrauss transforms and AMS sketches. Beyond sketching, our results uncover structural parallels to adversarial attacks in image classification, highlighting fundamental vulnerabilities of compressed representations.

## 1 Introduction

Dimensionality reduction is a fundamental technique in data analysis, algorithm design, and machine learning. A common paradigm is to apply a *sketching map*, a compressive transformation $C : \mathbb{R}^n \to \mathbb{R}^k$, which maps a high-dimensional input vector to a lower-dimensional representation. The map is typically sampled from a known distribution (e.g., The Johnson–Lindenstrauss (JL) transform [29] ) or learned during training, and is designed to support estimation of specific properties of the input $\boldsymbol{v}$, such as norms, inner products, or distances, using only the sketch $C(\boldsymbol{v})$.

Once constructed, the sketching map $C$ typically remains *fixed* across all inputs. This is true when compression layers are part of a trained model, and it is necessary in algorithmic contexts that require *composability*—the ability to sketch distributed datasets independently and combine the results—without revisiting the original data. A fixed sketching map also supports downstream tasks that operate directly in sketch space.

However, this compression introduces an intrinsic vulnerability: small input perturbations can produce large changes in the sketch, even when the true property (such as the norm) is nearly unchanged. For linear sketching maps $A \in \mathbb{R}^{k \times n}$, this vulnerability arises from structural facts such as the existence

of nontrivial null-space vectors and directions along which small-norm perturbations yield large distortions.

Such *adversarial inputs* can be constructed easily in the white-box setting, where the sketching map $C$ is known. In this work, we focus on the *black-box attack model*, which captures realistic situations where the adversary does not see the sketching map directly but can access it interactively via a *responder* using queries of the following form:

   (i) The adversary chooses a query vector $\boldsymbol{v} \in \mathbb{R}^n$.

   (ii) The system receives $\boldsymbol{v}$ and computes a sketch $C(\boldsymbol{v})$.

   (iii) The responder selects a map $\psi$ of sketches to distributions, receives the sketch $C(\boldsymbol{v})$ from system and returns a (possibly randomized) value $s \sim \psi(C(\boldsymbol{v}))$ to the adversary.

The goal of the adversary, in general terms, is to compromise the sketching map by causing a failure: If responses are correct, construct an adversarial input. We measure the efficiency of such *attacks* by the *number of queries* they require, as a function of the sketch size $k$.

We distinguish between two settings. In the *nonadaptive* case, all queries (including the final candidate) are chosen without regard to responses from earlier queries. The adversary performs a blind search, and success depends on the density of adversarial directions in the input space. In contrast, in the *adaptive* setting, each query can depend on prior responses, allowing the adversary to extract information about the sketching map and potentially converge to an adversarial input more efficiently.

The black-box model in the adaptive setting is well studied across multiple areas, including statistical queries [19, 28, 31, 25, 17, 6], sketching and streaming algorithms [33, 26, 9, 27, 40, 5, 8, 13, 15, 1, 22, 16], dynamic graph algorithms [36, 2, 20, 24, 39, 7], and machine learning [37, 21, 4, 34, 32, 35].

In this study, we aim to better understand this vulnerability for the task of $\ell_2$-norm estimation and to gain more general insights through this lens. Specifically, we consider linear transformations specified by a *sketching matrix* $A \in \mathbb{R}^{k \times n}$ which maps $\boldsymbol{v} \in \mathbb{R}^n$ to a *sketch* $A\boldsymbol{v} \in \mathbb{R}^k$.

Two classic methods for this task are the Johnson–Lindenstrauss (JL) transform [29] and the AMS sketch [3], They define *distributions* over sketching matrices that approximately preserve Euclidean norms (and therefore support sketch-based approximation of distances). The provided guarantees are probabilistic: for any input vector, with high probability over the random choice of the sketching matrix $A \in \mathbb{R}^{k \times n}$, the scaled norm of the sketch $\|A\boldsymbol{v}\|_2$ closely approximates $\|\boldsymbol{v}\|_2$ (within relative error $\epsilon$ with probability $1 - \delta$ when $k = O(\epsilon^{-2} \log(1/\delta))$).

However, as said, in practice, the sketching matrix is typically *fixed* over all input vectors, and importantly, no fixed matrix can preserve approximate norms for *all* inputs: every sketching matrix inevitably has inputs on which it fails. In the nonadaptive setting, the guarantees of JL and AMS apply via a union bound: with high probability (at least $1 - \delta$) over the sampling of matrices, the fixed sampled sketching matrix supports up to $\delta e^{O(k\epsilon^2)}$ approximate norm queries, in the sense that with probability $1 - \delta$, they are all accurate to within relative error $\epsilon$. Therefore, finding a bad input requires a number of queries that is exponential in $k$.

The adaptive setting was studied in multiple works. In terms of positive results, it is known that JL and AMS sketches can effectively trade off baseline (non-adaptive) accuracy and robustness within a fixed sketch size budget of $k$. By using carefully designed "robust" estimators [27, 10], the sketch can support for a fixed $\epsilon$, a number of adaptive queries that is *quadratic* in $k$. The idea, based on [18, 6], is to protect information on the sketching matrix by adding noise to the returned "best possible" estimate (or by subsampling a part of the sketch for each response).

As for negative results, Hardt and Woodruff [26] constructed an attack of size polynomial in $k$ that for any sketching matrix constructs a distribution over inputs under which any estimator would fail. The idea in the attack is to identify vectors that lie close to the null space of $A$ and hence are transparent to the sketch. Cherapanamjeri and Nelson [12], Ben-Eliezer et al. [9] presented an attack of size linear in $k$ on the JL and AMS sketches with the standard estimator (which returns a scaled norm of the sketch). The product of these attack is a vector on which the standard estimator fails. Cohen et al. [15] constructed an attack of quadratic size on the AMS sketch that is *universal* (applies against any query responder), that constructs a vector on which the standard estimator fails. These negative results vary by (i) the scope of the sketching matrices compromised (general or JL/AMS), (ii) the

power of the query responder (strategic and adaptive or the standard estimator), and (iii) the product of the attack (a distribution that fails any query responder or a single vector with an out-of-distribution sketch that fails the "optimal" estimator).

An intriguing gap remains, however, between the established quadratic guarantee for the number of adaptive queries with correct responses and the super-quadratic sizes of the known attacks for general sketching matrices.

## 1.1 Overview of contributions

Our primary contribution in this work, is a construction of an attack of quadratic size in $k$ that applies against *any* sketching matrix $A \in \mathbb{R}^{k \times n}$ and with any query responder for $\ell_2$ norm estimation. The attack produces a vector with an out-of-distribution sketch:

**Theorem 1.1** (Attack Properties). *There exist a universal constant $C > 0$, and families of distributions $\mathcal{F}_n$ over $\mathbb{R}^n$, such that the following holds.*

*For every sketching matrix $A \in \mathbb{R}^{k \times n}$ with $n = \Omega(k^2)$, any query responder, and deviation $\gamma \geq 1$ and accuracy $\alpha \in (0, 1)$ parameters, with probability at least $0.9$ over the choice of a distribution $\mathcal{D} \sim \mathcal{F}$, after $r = C\gamma^2\alpha^2k^2 \log^2 k$ i.i.d. queries $\boldsymbol{v} \sim \mathcal{D}$, one of the following outcomes occurs:*

   *(i) At least $\delta(\alpha) > 0$[1] fraction of the responses have relative error greater than $\alpha$, or*

   *(ii) We construct a query vector on which the optimal estimator (with respect to $A$ and $\mathcal{D}$) returns a value that deviates from the true norm by at least a multiplicative factor of $\gamma$.*

Our attack deploys a simple and natural query distribution that combines a weighted sparse signal with additive dense noise:
$$\boldsymbol{v} = w\boldsymbol{e}_h + \boldsymbol{u},$$
where $\boldsymbol{e}_h$ is the standard basis vector corresponding to index $h \in [n]$, and $\boldsymbol{u}$ is Gaussian noise supported on a set $M \subset [n] \setminus \{h\}$ of size $m > k^2$.

Importantly, sparse queries alone are insufficient for attack: under the mild assumption that the sketching matrix $A$ has its columns in general position, the $\ell_2$ norm of any $k$-sparse vector can be exactly recovered from its sketch (e.g., [11]). This implies that if the attack is restricted to sparse inputs, the response leaks no information about the sketching matrix and full robustness is preserved.

To sample from our query distribution, we first choose the *signal coordinate* $h \sim [n]$ uniformly at random, and then independently sample a *noise support* $M \subset [n] \setminus \{h\}$ of size $m = \Omega(k^2)$.

The goal of the query responder is to estimate the $\ell_2$ norm of $\boldsymbol{v}$ from its sketch $A\boldsymbol{v}$, with relative error at most $\alpha$. Our attack remains effective even against the simpler *norm gap* task: return $-1$ if $\|\boldsymbol{v}\|_2 \leq 1$ and return $1$ if $\|\boldsymbol{v}\|_2 \geq 1 + \alpha$, with either output permitted when $\|\boldsymbol{v}\|_2 \in (1, 1 + \alpha)$.

Note that the norm gap task reveals strictly less information than norm estimation: a norm estimate with relative error $\Theta(\alpha)$ trivially yields a correct solution to the norm gap task with parameter $\Theta(\alpha)$, but not vice versa.

Our attack is described in Algorithm 1. Query vectors $(\boldsymbol{v}^{(t)})_{t \in [r]}$ are constructed by sampling a signal value $w^{(t)} \sim W$, where $W$ is a probability distribution over $\mathbb{R}$, and Gaussian noise $\boldsymbol{u}^{(t)}$ to form $\boldsymbol{v}^{(t)} = w^{(t)}\boldsymbol{e}_h + \boldsymbol{u}^{(t)}$. The adversary collects the responses $s^{(t)}$ for the sketch $A\boldsymbol{v}^{(t)}$. We establish that if the responses are correct for the norm gap (except for a small fraction of queries) then the normalized signed sum of the noise vectors $\sum_t s^{(t)}\boldsymbol{u}^{(t)}$ is adversarial.

**Universality and Limitations**   Our attack is *universal* in that it applies with *any* query responder. The analysis allows the responder to be strategic and adaptive, with full knowledge of the query distribution $\mathcal{D}$ and the internal state of the attacker. Notably, the attack is *single batch* – it uses adaptivity in the minimal possible way: all queries are generated independently and only the final adversarial vector is constructed adaptively from the responses.[2]

---

[1]A constant that depends on $\alpha$

[2]Single batch attacks were constructed in prior works for JL, Count-Sketch, AMS, and Cardinality sketches [12–14, 1].

A limitation of our result is that the attack guarantees failure only for the *optimal estimator* tailored to $\mathcal{D}$ and $A$, rather than for *every* possible query responder. The stronger goal—constructing a distribution over inputs that defeats all responders with high probability—remains open.

Despite this, we believe our result is both theoretically meaningful and practically relevant. Theoretically, we obtain a *tight* quadratic bound in the batch-query model, whereas known attacks with the stronger guarantee require significantly more queries (a higher degree polynomial in $k$). Moreover, any attack achieving the stronger guarantee would require at least $\tilde{\Omega}(k)$ *adaptive batches*, and thus cannot be realized within our single-batch setting. In this sense, the adversarial vector we construct is the strongest outcome achievable in our setting.

Our attack product is practically relevant because the optimization process in a model training tends to converge to an (at least locally) optimal estimator for the training distribution. Therefore, compromising the model means compromising this specific implemented estimator, coded in the model parameter values, rather than any possible query responder.

**Roadmap** Our attack algorithm is described in Section 3 with the analysis presented in Section 4. An empirical study of our attack on JL and AMS sketching matrices is included in Section 5. We conclude in Section 6 with a discussion of open directions and implications to image classifiers.

## 2 Preliminaries

We denote vectors in boldface $\boldsymbol{v}$, scalars $v$ in non boldface, and inner product of two vectors by $\langle \boldsymbol{v}, \boldsymbol{u} \rangle = \sum_i v_i u_i$. For a vector $\boldsymbol{v} \in \mathbb{R}^n$, we refer to $i \in [n]$ as a *key* and to $v_i$ as the value of the $i$th key. For $M \subset [n]$ let $v_M$ be the projection of $\boldsymbol{v}$ on entries $(v_i)_{i \in M}$. To streamline the presentation, we interchangeably use the same notation to refer to both a random variable and a distribution.

$\mathcal{N}(\mu, \sigma^2)$ is the normal distribution with mean 0 and variance $\sigma^2$ with density function

$$\varphi_{\mu, \sigma^2}(x) := \frac{1}{\sigma\sqrt{2\pi}} e^{-\frac{1}{2}\left(\frac{x-\mu}{\sigma}\right)^2} . \tag{1}$$

$\mathcal{N}_\ell(0, \sigma^2)$ is the $\ell$-dimensional Gaussian distribution with covariance matrix $I_\ell$ ($\ell$ i.i.d. $\mathcal{N}(0, \sigma^2)$). Its probability density function is

$$f_\sigma(\boldsymbol{u}) = \frac{1}{(\sigma\sqrt{2\pi})^\ell} e^{-\frac{\|\boldsymbol{u}\|_2^2}{2\sigma^2}} . \tag{2}$$

A *linear sketching map* is defined by a *sketching matrix* $A \in \mathbb{R}^{k \times n}$ where $k \ll n$. The input is represented as a vector $\boldsymbol{v} \in \mathbb{R}^n$ and the *sketch* of $\boldsymbol{v}$ is the product $A\boldsymbol{v} \in \mathbb{R}^k$.

## 3 Attack description

**Definition 3.1** (($y, \alpha$)-Gap Problem). Given width parameter $\alpha > 0$ and $y \in \mathbb{R}$, a *gap problem* is, for input $x \in \mathbb{R}$ to return $-1$ when $x \leq y$ and to return 1 when $x \geq y + \alpha$. The output may be arbitrary in $\{-1, 1\}$ if $\|\boldsymbol{v}\|_2 \in (y, y + \alpha)$.

Observe that an additive approximation of $\alpha/2$ or a multiplicative approximation of $\alpha/(2(y + \alpha)$ for $x$ yields a correct solution to the $(\ell, \alpha)$-gap problem for $x$. Hence an attack that is effective with the weaker norm gap responses is more powerful.

Our attack is described in Algorithm 1. The signal density and parameter settings are described in Definition 3.3.

In each of $r$ attack steps,

1. The adversary samples an independent query vector $\boldsymbol{v} := w\boldsymbol{e}_h + c\boldsymbol{z}$ by sampling Gaussian noise with support $M$ $\boldsymbol{z} \sim \mathcal{N}_{n,M}(0, \sigma^2 = 1/m)$, sampling signal weight $w \sim W$.

2. The responder chooses an estimator map $\psi : \mathbb{R}^k \to \mathcal{P}(\{-1, 1\})$, obtains the sketch $A\boldsymbol{v}$ from the system, and returns $s \sim \psi(A\boldsymbol{v})$ to the adversary.

**Algorithm 1:** Universal Attack on Sketching Matrix $A$; $\ell_2$ norm gap responder

---

**Input:** $A \in \mathbb{R}^{k \times n}$, accuracy parameter $\alpha$, number of queries $r$, signal index $h \in [n]$, support $M \subset [n]$ of size $m = |M|$, noise scale factor $c$

**for** $t \in [r]$ **do** // Main loop

$\quad z_i^{(t)} \sim \begin{cases} 0 & \text{if } i \notin M \\ \mathcal{N}(0, 1/m) & \text{if } i \in M \end{cases}$      // sample noise vector

$\quad w^{(t)} \sim W$      // Sample signal weight from $W$ (Definition 3.3)

$\quad \boldsymbol{v}^{(t)} = w^{(t)} \cdot \boldsymbol{e}_h + c\boldsymbol{z}^{(t)}$      // Query vector

$\quad$ Responder chooses an estimator $\psi^{(t)} : \mathbb{R}^k \to \mathcal{P}\{-1, 1\}$      // Map sketches to responses

$\quad s^{(t)} \sim \psi^{(t)}(A\boldsymbol{v}^{(t)})$      // Responder receives sketch, returns $s^{(t)} \in \{-1, 1\}$

**return** $\boldsymbol{z}^{(\text{adv})} \leftarrow \dfrac{\sum_{t \in [r]} s^{(t)} \boldsymbol{z}^{(t)}}{\|\sum_{t \in [r]} s^{(t)} \boldsymbol{z}^{(t)}\|_2}$      // Adversarial noise

---

    3. The adversary then adds $s\boldsymbol{z}$ to the accumulated output.

The product of the attack $\boldsymbol{z}^{(\text{adv})}$ is the normalized accumulated output.

**Theorem 3.2** (Attack efficacy). *Let $A \in \mathbb{R}^{k \times n}$ be a sketching matrix. Consider applying Algorithm 1 with a randomly selected $h \in [n]$ and support $M \subset [n] \setminus \{h\}$ of size $m = \Omega(k^2)$ and $r = O(\gamma^2 \alpha^2 k^2 \log^2 k)$ queries. Then with constant probability one of the following holds. Either the error rate of responses $s^{(t)}$ for the $(1, \alpha)$-gap problem on the input norm $\|\boldsymbol{v}^{(t)}\|_2$ (see Definition 3.1) exceeded some constant $\delta(\alpha) > 0$ or the vector $\boldsymbol{z}^{(\text{adv})}$ is a $\gamma$-adversarial noise vector (see Definition 4.3) for $A$, signal $h$ and noise support $M$.*

We define a vector as $\gamma$-*adversarial* if, under the query distribution specified by $h$ and $M$, the optimal estimator returns a value that deviates from the true norm by a multiplicative factor of at least $\gamma$.

## 3.1 Density and parameter setting

**Definition 3.3** (Signal density and parameters). The distribution $W$ we use for $w$ is parametrized by $a < 1 < 1 + \alpha < b$ and has density function:

$$C = \frac{2}{b - a + \alpha}, \qquad \nu(w) = \begin{cases} 0, & w < a \text{ or } w > b, \\ C\,\dfrac{w - a}{1 - a}, & a \leq w \leq 1, \\ C, & 1 < w < 1 + \alpha, \\ C\,\dfrac{b - w}{b - 1 - \alpha}, & 1 + \alpha \leq w \leq b. \end{cases}$$

We set the parameters as follows according to the gap $\alpha > 0$ and the error rate $\delta > 0$ allowed for the responder. We use $\delta > 0$ that satisfies $\delta / \log(1/\delta) = O(\alpha^2)$ and $c$ that is a small constant (that does not depend on $\alpha$ and is selected according to other constants).

$$a = 1 - 10\alpha/c$$
$$b = 1 + \alpha + 10\alpha/c$$

Observe from our settings that each of the intervals $[a, 1]$ and $[1 + \alpha, b]$ has at least a constant fraction of the probability mass.

## 4 Analysis of the attack

This section presents the key components and outlines the proof of Theorem 3.2, with full details deferred to the appendix.

We introduce notation for the query and noise distributions. For indices $M \subset [n]$, let $\mathcal{N}_{n,M}$ be the distribution over $\boldsymbol{u} \in \mathbb{R}^n$ in which $u_{[n] \setminus M} = \boldsymbol{0}$ and the coordinates indexed by $M$ are sampled from the $m = |M|$-dimensional Gaussian distribution $u_M \sim \mathcal{N}(0, \frac{1}{m} I_m)$. Note that $\mathcal{N}_{n,M}$ is the distribution of noise vectors selected in Algorithm 1.

For $h \in [n]$, $M \subset [n] \setminus \{h\}$, and noise scale $c$ let

$$F_{h,M,c}[w] := w\boldsymbol{e}_h + c\mathcal{N}_{n,M} \tag{3}$$

be the distribution of vectors formed by adding a scaled noise vector sampled from $\mathcal{N}_{n,M}$ to a *signal* $w\boldsymbol{e}_h$, where $\boldsymbol{e}_h \in \mathbb{R}^n$ is the standard basis vector at index $h$. $F_{h,M,c}[w]$ is the distribution of query vectors selected in Algorithm 1 for signal value $w$.

Our analysis is in terms of *signal estimation*. In order to facilitate it, we establish that a correct norm gap output yields a correct signal gap output with similar parameters:

**Lemma 4.1** (Norm gap to signal gap). *With the choice of parameters for our attack and $m = \Omega((k+r)\log((k+r)/\delta))$, with probability close to 1, a correct $(1-c^2, 1.1\alpha)$-norm gap output implies a correct $(1,\alpha)$-signal gap output on all queries.*

The proof is included in Appendix A. The norm gap in the statement of Theorem 3.2 is with parameters $(1,\alpha)$. To reduce clutter, we treat it in the sequel as signal gap of $(1,\alpha)$.

## 4.1 Signal estimation and the optimal estimator

For a fixed sketching matrix $A$, $h \in [n]$, and noise support $M$, we express the optimal estimator on the signal $w$ from a sketch $A\boldsymbol{v}$ when $\boldsymbol{v} \sim F_{h,M,c}[w]$. For this purpose we may assume that the distributions (and $A$, $h$, and $M$) are given to the responder. Note that if it holds that $A_{\bullet h} = \boldsymbol{0}$, then the sketch carries no information on the signal $w$. When this is not the case, we can express the optimal unbiased estimator. The proof is included in Appendix B.

**Lemma 4.2** (Estimator for the Signal). *Fix $h \in [n]$, a noise support set $M \subset [n] \setminus \{h\}$, and a noise scale factor $c$. Consider the distributions $F_{h,M,c}[w]$ parametrized by $w$.*

*If the column $A_{\bullet h}$ is nonzero, then there exists an unbiased, complete, and sufficient statistic $T_{h,M} : \mathbb{R}^k \to \mathbb{R}$ for the signal $w$ based on the sketch $A\boldsymbol{v}$.*

*Furthermore, the deviation of this estimator from its mean, defined as*

$$\Delta_{h,M}(cA\boldsymbol{u}) := T_{h,M}(A\boldsymbol{v}) - w,$$

*depends only on the sketch of the noise $\boldsymbol{u} \sim \mathcal{N}_{n,M}$, and is distributed as a Gaussian random variable $\mathcal{N}(0, c^2 \sigma_T^2(h, M))$.*

The estimator $T_{h,M}(A\boldsymbol{v})$ also minimizes the mean squared error (MSE) [30].

We define an adversarial noise vector to be one that causes a large deviation in this optimal estimator:

**Definition 4.3** ($\gamma$ adversarial noise). *A unit vector $\boldsymbol{u}$ with support $M$ is $\gamma$-adversarial for $A$, $h$, $M$ if $|\Delta_{h,M}(\boldsymbol{u})| > \gamma$*

Adversity of $\gamma$ means that the value is $\gamma/(c\sigma_T)$ standard deviations off. We will see that the attack size needed for certain adversity depends on $\sigma_T$.

## 4.2 Lower bounding the error

Since $T_{h,M}(A\boldsymbol{v})$ minimizes the MSE for estimating the signal $w$ from the sketch, it implies a lower bound on the error that applies for any query responder on the query distribution of Algorithm 1:

**Corollary 4.4.** The mean squared error (MSE) on *any* estimator, even one that is tailored to $h$, $M$, and $c$, on queries of the form $\boldsymbol{v} \sim F_{h,M,c}[w]$ when $w \sim W$ is $\Omega(c^2 \sigma_T(h, M)^2)$.

We next establish that for any sketching matrix $A \in \mathbb{R}^{k \times n}$, a random choice of signal index $h \in [n]$ and noise support (of size $M$ that is slightly superlinear in $k$), it is likely that either column $h$ is all zeros (and hence any estimator must fail with constant probability) or $\sigma_T^2(h, M) = \tilde{\Omega}(1/k)$. The proof is in Appendix C.

**Lemma 4.5** (Lower Bound on Error). *Let $A \in \mathbb{R}^{k \times (m+1)}$ be a matrix with $m \geq 20k \log^2 k$. Then, for at least $0.9$ fraction of columns $h \in [m+1]$, it holds that either $A_{\bullet h}$ is all zeros or $\sigma_T^2(h, M = [m+1] \setminus \{h\}) = \Omega\left(\frac{1}{k \log k}\right)$.*

Recall that when the input vectors are $k$-sparse, exact recovery of the norm is possible, and hence there is no estimation error. Therefore, there is potential vulnerability only when the sparsity of the query vectors exceeds $k$ and thus our slightly super linear sparsity is necessary.

As a corollary of Lemma 4.5, we obtain a lower bound on the relative error guarantees of any estimator:

**Corollary 4.6.** Consider a fixed sketching matrix Let $A \in \mathbb{R}^{k \times n}$ be a sketching matrix with $n = \Omega(k \log^2 k)$. Consider the distribution of queries that samples $h$ and $M$ (of size $m = \Omega(k \log^2 k)$) randomly from $[n]$ and then sample a query from $F_{h,M,c}[w]$ where $w \sim W$ (and the density of $W$ is at least a constant in range of size $> 100/\sqrt{k}$). Then with at least constant probability, the MSE of any sketch-based signal (and hence norm) estimator is $\Omega(1/(k \log k))$.

**Remark 4.7** (Error for JL matrices). Sampled matrices from the JL distributions meet this upper bound: For fixed noise support $M$ of size $m > k \log k$, with high probability over the sampling of $A$, for all $h \in [n]$, $\sigma_T^2(h, M) = O(1/k)$ (the property needed for the sampled $A$ is that the projected rows $A_{iM}$ are close to orthogonal).

## 4.3 The gain lemma

We quantify the expected progress (towards an adversarial vector) in each step of the attack in terms of $\sigma_T^2(h, M)$. What we show is that when the estimator has a low error rate, then queries for which the noise component $\boldsymbol{u}$ has a higher deviation $\Delta_{h,M}(A\boldsymbol{u})$ are more likely to have $s^{(t)} = 1$. Specifically, even though $\mathsf{E}[\Delta_{h,M}(A\boldsymbol{u})] = 0$ (as the deviation has distribution $\mathcal{N}(0, \sigma_T^2)$), the response $s^{(t)}$ is correlated with it and we will show that $\mathsf{E}[\Delta_{h,M}(s^{(t)} A\boldsymbol{u})] \propto \sigma_T^2(h, M)$.

**Definition 4.8** (Error rate of sketch-based signal gap estimator). An estimator $\psi : \mathbb{R}^k \to \mathcal{P}\{-1, 1\}$ is a map from a sketch to a probability distribution on $\{-1, 1\}$. For a sketching map $A \in \mathbb{R}^{k \times n}$, $h, M, c$ and signal distribution $W$, the *error rate* of $\psi$ for the $\alpha$-signal gap problem (Definition 3.1) is the probability over the query distribution of an incorrect $(1, \alpha)$- signal gap response:
$$\mathsf{err}(\psi) := \mathop{\mathsf{E}}_{w \sim W, \boldsymbol{v} \sim F_{h,M,c}[w]} [\Pr[\psi(A\boldsymbol{v}) = 1] \cdot \mathbf{1}\{w \leq 1\} + \Pr[\psi(A\boldsymbol{v}) = -1] \cdot \mathbf{1}\{w \geq 1 + \alpha\}] .$$

If $A_{\bullet h} = \mathbf{0}$ then any estimator would have at least a constant error rate. Additionally, from standard tail bounds, for error rate that is $O(\delta)$, it must hold that $\alpha > c\,\Omega(\sqrt{\log(1/\delta)})\sigma_T$ and therefore $\sigma_T < \alpha/(c\sqrt{\log(1/\delta)})$.

We therefore consider the case where $A_{\bullet h} \neq \mathbf{0}$, and therefore unbiased $T_{h,M}$ exists and $\sigma_T^2(h, M)$ is well defined, and assume that $\sigma_T < \alpha/(c\sqrt{\log(1/\delta)})$. We quantify the per-step expected gain (see Appendix D for the proof):

**Lemma 4.9** (Gain Lemma). *If $\mathsf{err}[\psi] < \delta$ and attack parameters are as in Definition 3.3, and $\sigma_T < \alpha/(c\sqrt{\log(1/\delta)})$ then*
$$\mathop{\mathsf{E}}_{\boldsymbol{v}} [\psi(A\boldsymbol{v})(T_{h,M}(A\boldsymbol{v}) - w)] = \Omega(\alpha^{-1} c^3 \sigma_T(h, M)^2)$$

## 4.4 Proof of Attack efficacy Theorem

*Proof of Theorem 3.2.* The proof idea is to bound the deviation and the norm of the sum $\sum_{t \in [r]} s^{(t)} \boldsymbol{z}^{(t)}$ and show that the deviation increases faster than the norm.

We first bound norm. We show that if the noise support is sufficiently wide with $m > r^2$, then for *any* choice of $s^{(t)}$, the norm of $\sum_{t \in [r]} s^{(t)} \boldsymbol{z}^{(t)}$ can not be much higher than that of the sum $\sum_{t \in [r]} \boldsymbol{z}^{(t)}$ of independent Gaussians:
$$\left\| \sum_{t \in [r]} s^{(t)} \boldsymbol{z}^{(t)} \right\|_2 = O(\sqrt{r}) \tag{4}$$

This follows as an immediate corollary of Lemma E.1.

We now consider the deviation of the sum. From Lemma 4.9 we obtain that for each $t$, $\mathsf{E}_{\boldsymbol{v}}[s^{(t)}\Delta_{h,M}(A\boldsymbol{z}^{(t)})] = \Omega(\alpha^{-1}\sigma_T^2(h,M))$.

From concentration of $\Delta_{h,M}(A\boldsymbol{z}^{(t)})]$ (that are independent $\mathcal{N}(0,\sigma_T^2)$) we obtain that with high probability

$$\sum_{i=1}^{r} s^{(t)}\Delta_{h,M}(A\boldsymbol{z}^{(t)}) = \Omega(r\,\alpha^{-1}\,\sigma_T^2(h,M))\,. \tag{5}$$

From Lemma 4.5, for any sketching matrix $A \in \mathbb{R}^{k \times n}$, when we sample $h$, and $M \subset [n]$ of size $m = \Omega(k\log^2 k)$, then with constant probability we have $\sigma_T^2(h,M) = \Omega(\frac{1}{k\log k})$.

Combining with (5) we obtain that

$$\sum_{i=1}^{r} s^{(t)}\Delta_{h,M}(A\boldsymbol{z}^{(t)}) = \Omega\left(\frac{r\alpha^{-1}}{k\log k}\right)\,. \tag{6}$$

Combining (4) and (6) we obtain that with constant probability

$$\Delta_{h,M}(A\boldsymbol{z}^{(\mathrm{adv})}) = \Omega\left(\frac{\sqrt{r}\alpha^{-1}}{k\log k}\right)\,.$$

For the deviation to exceed $\gamma$, solving for $\frac{\sqrt{r}\alpha^{-1}}{k\log k} > \gamma$, we obtain $r = O(\gamma^2\,\alpha^2\,k^2\log^2 k)$.

This concludes the proof. $\qquad\square$

## 5  Empirical study

We implemented Algorithm 1 and evaluated its effectiveness on two families of sketching matrices: Gaussian Johnson–Lindenstrauss (JL) transforms [29] with $A \in \mathbb{R}^{k \times n}$ with i.i.d. entries $A_{ij} \sim \mathcal{N}(0,1/k)$ and AMS sketches [3] with $A \in \{\pm 1\}^{k \times n}$ consisting of i.i.d. Rademacher entries ($A_{ij} = \pm 1$ with equal probability). Our evaluation used different configurations of sketch dimension $k$ and ambient dimension $n$ (effectively, the noise support). For each configuration we applied the attack against the corresponding *standard* norm estimator and its *robustified* variant, and measured how rapidly the adversarial bias grows.

**Standard Estimators.**   For Gaussian JL matrices, the *standard* norm estimator is simply the sketch norm, which coincides with the minimum-variance unbiased estimator for $\|v\|_2^2$ under Gaussian projections:

$$\widehat{\|v\|_2^2} := \|A\boldsymbol{v}\|_2^2.$$

For AMS sketches, we implemented the *median-of-means* (MoM) estimator. Specifically, we partition the $k$ rows of $A$ into $g = \max\{5, \lfloor\sqrt{k}\rfloor\}$ groups of equal size $b = \lceil k/g\rceil$, compute the mean of $y_i^2$ within each group, and take the median of the $g$ group means:

$$\widehat{\|v\|_2^2} := \mathrm{median}\left(\frac{1}{b}\sum_{i\in G_1} y_i^2,\ \frac{1}{b}\sum_{i\in G_2} y_i^2,\ \ldots,\ \frac{1}{b}\sum_{i\in G_g} y_i^2\right).$$

Our final estimate is $\widehat{\|v\|_2} := \sqrt{\widehat{\|v\|_2^2}}$.

**Simplified attack**   We implemented a simplified attack that sufficed for the special case of JL/AMS matrices (distribution that is invariant under column permutations) and query responders that are not adaptive and not tailored to the input distribution). Our query vectors are sampled i.i.d. from $F_{n,[n-1],1}[w=1]$ (see Eq. (3)), that is, use a a fixed signal value $w=1$ and fixed $h=n$ and $M = [n-1]$. Observe that for our query vectors, the norm $\|v\|_2$ is concentrated around $\sqrt{w^2+1} = \sqrt{2}$.

**Robust Estimators.** The robust variants are parameterized by a noise scale $\sigma$ and add Gaussian noise to the squared-norm standard estimate before taking the square root:

$$\hat{s}_\sigma(\boldsymbol{v}) := \sqrt{\widehat{\|v\|_2^2} + \mathcal{N}(0, \sigma^2)}.$$

In our attack implementation the responder outputs $s = 1$ when $\hat{s}_\sigma \geq \sqrt{w^2 + 1}$ and $s = -1$ otherwise. [3]

On freshly sampled (non-adaptive) inputs, the estimator has standard deviation $1/\sqrt{k}$ in the JL case, whereas for the robustified estimator the standard deviation increases to $\sqrt{\frac{1}{k} + \sigma^2}$. Therefore, the non-adaptive accuracy decreases with the robustness noise $\sigma$.

**Experiments** For each $(k, n)$ configuration of sketch size $k$ and support size $n$, we performed rep $=$ 20 trials. For each trial we sample a fresh sketching matrix $A \in \mathbb{R}^{k \times n}$. We sample query vectors as described and apply estimators with different values of the robustness noise $\sigma \in \{0, 0.1, 0.2, 0.4\}$ to the same query vectors (observe that $\sigma = 0$ is the standard estimator). We study the effectiveness of our attack by tracking the ratio $\hat{s}_\sigma(\boldsymbol{z}^{(\mathrm{adv})})/\|\boldsymbol{z}^{(\mathrm{adv})}\|_2$, where $\hat{s}_\sigma$ is the (respective) standard estimator that is applied to the sketch and $\boldsymbol{z}^{(\mathrm{adv})}$ is the adversarial vector.

**Results** Fig. 1 and Fig. 2 report, for different configurations with $k \in \{100, 250, 1000\}$, the ratio $\hat{s}_\sigma(\boldsymbol{z}^{(\mathrm{adv})})/\|\boldsymbol{z}^{(\mathrm{adv})}\|_2$ (the $y$-axis) as a function of the number of attack queries (the $x$-axis).

We observe that the attack is effective even in configurations with small support size of $n/k \in [2, 5]$. The standard and lower-$\sigma$ robust estimators are initially more accurate but also accumulate bias faster and incur a much higher error as the attack progresses. A larger sketch size $k$ results in higher robustness (slower increase in the bias) for the same noise $\sigma$. The plots show a quadratic pattern where the bias induced by the attack increases like square root of the number of queries. Results for additional configurations for the same $k$ are reported in Fig. 3. We observe that the effectiveness of the attack tends to increase with a larger support size $n$.

**Discussion.** Our empirical results demonstrate that the attack is substantially more effective in practice than our current analysis predicts. In particular, we observe strong empirical success even when the noise support size is very small—on the order of $O(k)$—whereas our analysis requires $n = \Omega(k^2)$. The attack is also effective with relatively few queries, again beyond what is guaranteed by our theoretical bounds.

Moreover, our analysis only establishes that the attack product compromises the *optimal* estimator for the sketching matrix, which is a linear statistic of the sketch. For JL sketches, the standard estimator is very close (up to row orthogonality) to this optimal estimator, so our theory is largely predictive. In contrast, for AMS sketches we employ a median-of-means estimator, which is non-linear, yet the attack empirically compromises it with comparable efficiency. This suggests that the vulnerability extends beyond linear estimators and may be more fundamental than our current proofs capture.

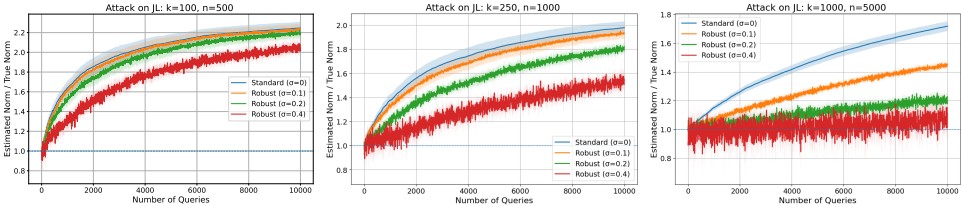

Figure 1: Attack on JL with $(k, n) \in \{(100, 500), (250, 1000), (1000, 5000)\}$. Standard and robust estimators. Mean ratio of estimate to actual norm with 95% confidence intervals over 20 repetitions.

---

[3]Our simplified attack would fail against a fully strategic responder that is tuned to the input distribution and always returns the deterministic value $\sqrt{w^2 + 1} = \sqrt{2}$. Such a responder leaks no information about the sketching matrix and is correct with high probability on all queries.

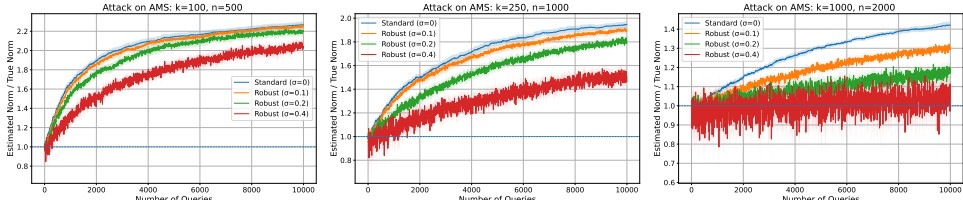

Figure 2: Attack on AMS with $(k, n) \in \{(100, 500), (250, 1000), (1000, 2000)\}$. Standard and robust estimators. Mean ratio of estimate to actual norm with 95% confidence intervals over 20 repetitions.

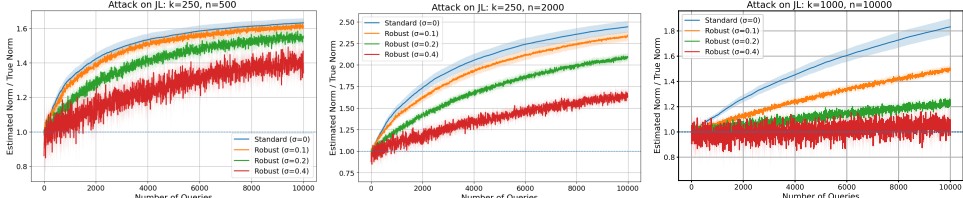

Figure 3: Attack on JL with $(k, n) \in \{(250, 500), (250, 2000), (1000, 10000)\}$. Mean ratio of estimate to actual norm with 95% confidence intervals over 20 repetitions.

## 6  Conclusion

Our results further suggest that vulnerability to black-box attacks is an inherent consequence of dimensionality reduction. Nonetheless, it can be partly mitigated by sacrificing some non-adaptive accuracy in exchange for increased robustness, underscoring a fundamental trade-off. We close with several directions for further investigation.

**Theory questions.** We proved a tight, quadratic-size universal attack that succeeds against any query responder, but our analysis only guarantees compromise of the *optimal* (linear) estimator for the sketch and attack distribution. Empirically we observed that our attack compromised also the non-linear median-of-means estimator for AMS. This suggests exploring broader applicability of our attack. Another natural question is whether our technique can be extended to other norms, as in Gribelyuk et al. [23]. A more ambitious goal is to construct a quadratic-size attack that generates a *single distribution* capable of compromising *every* query responder. Such an attack would necessarily require $\tilde{\Omega}(k)$ adaptive batches and could plausibly build on an enhanced version of our construction that incorporates the discovered adversarial directions into the input distribution [16]. Another open question concerns the required size of the noise support, which directly corresponds to the attacker's storage cost. Our current proof requires $n = \Omega(k^2)$ because it relies on a general bound (Lemma E.1) for arbitrary signed sums of Gaussian vectors. Empirically, however, we observe that our attack remains effective with support as small as $2k$, and we conjecture that $\tilde{O}(k)$ support should suffice.

**Connections to Image Classifiers.** Our findings on the vulnerability of linear sketches may shed light on the phenomenon of adversarial examples in image classification, which has been extensively studied in prior work [37, 21, 4, 34, 32, 35]. Many of these attacks exhibit a striking structural similarity to our setting: they construct adversarial examples by aggregating small, randomly oriented perturbations that consistently push the system's output in a particular direction. When normalized and accumulated, these perturbations yield a large, adversarial deviation that fools the model. This additive, alignment-based mechanism appears to be remarkably effective across domains.

Although image classifiers such as CNNs are highly nonlinear, they contain significant linear components, particularly in early layers. This observation raises a natural question: could adversarial susceptibility in image models arise, at least in part, from the same linearity-driven accumulation of aligned noise? If so, it may be possible to improve robustness by introducing carefully designed randomness—such as injecting noise that is unknown to the attacker into internal representations or into the input.

## Acknowledgments

Edith Cohen was partially supported by Israel Science Foundation (grant 1156/23). Uri Stemmer was Partially supported by the Israel Science Foundation (grant 1419/24) and the Blavatnik Family foundation.

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

## A  Norm estimation to signal estimation

This section contains a restatement and proof of Lemma 4.1:

**Lemma 4.1** (Norm gap to signal gap). *With the choice of parameters for our attack and $m = \Omega((k + r)\log((k + r)/\delta))$, with probability close to $1$, a correct $(1 - c^2, 1.1\alpha)$-norm gap output implies a correct $(1, \alpha)$-signal gap output on all queries.*

*Proof.* The query vectors $\boldsymbol{v} = w\boldsymbol{e}_h + c\boldsymbol{u}$ sampled in Algorithm 1 are from distributions $F_{h,M,c}[w]$, for $w \in [a, b]$. The squared norm is $\|\boldsymbol{v}\|_2^2 = w^2 + c^2\|\boldsymbol{u}\|_2^2$. We show that $\|\boldsymbol{u}\|_2^2$ is tightly concentrated around its expectation $1$:

**Claim A.1** (Concentration of $\|\boldsymbol{u}\|_2^2$).

$$\Pr\left[\left|\|\boldsymbol{u}\|_2^2 - 1\right| \geq \epsilon\right] \leq 2e^{-\frac{m}{2}(\epsilon^2/2)}$$

*Proof.* The sum of the squares of $m$ i.i.d. $\mathcal{N}(0, \sigma^2)$ has distribution $\sigma^2\chi_m^2$ and expected value $m\sigma^2$. Applying tail bounds on $\chi_m^2$ (Gaussian concentration of measure) we obtain that for $\epsilon \in (0, 1)$

$$\Pr\left[\frac{\left|\|\boldsymbol{u}\|_2^2 - m\sigma^2\right|}{m\sigma^2} \geq \epsilon\right] \leq 2e^{-\frac{m}{2}(\epsilon^2/2)} .$$

Substituting $\sigma^2 = 1/m$ we obtain the claim. $\qquad\square$

It follows that if we choose $m = \Omega((k + r)\log((k + r)/\delta))$ then with probability $1 - \delta$, on all our queries, the squared norm of the noise is within $\|\boldsymbol{u}\|_2^2 \in (1 \pm 1/(10\sqrt{k})$. Therefore, a correct $(1, \alpha)$-gap output on the norm yields a correct $(1, \alpha^2 + 2\alpha)$-gap on the squared norm $\|\boldsymbol{v}\|_2^2$. This gives an $(1 - c^2(1 + 1/(10\sqrt{k})), \alpha^2 + 2\alpha - c^2/(10\sqrt{k}))$-gap output on the squared signal $w^2$.

We now note that we can assume $\alpha > 1/\sqrt{k}$ because otherwise, any responder would be incorrect with constant probability. Using our parameter setting of $\ell$ close to $1$ and fixed $c \ll 1$ we obtain the claim in the statement of the lemma. $\qquad\square$

## B  Estimator for the Signal

This section contains a restatement and proof of Lemma 4.2.

**Lemma 4.2** (Estimator for the Signal). *Fix $h \in [n]$, a noise support set $M \subset [n] \setminus \{h\}$, and a noise scale factor $c$. Consider the distributions $F_{h,M,c}[w]$ parametrized by $w$.*

*If the column $A_{\bullet h}$ is nonzero, then there exists an unbiased, complete, and sufficient statistic $T_{h,M} : \mathbb{R}^k \to \mathbb{R}$ for the signal $w$ based on the sketch $A\boldsymbol{v}$.*

*Furthermore, the deviation of this estimator from its mean, defined as*

$$\Delta_{h,M}(cA\boldsymbol{u}) := T_{h,M}(A\boldsymbol{v}) - w,$$

*depends only on the sketch of the noise $\boldsymbol{u} \sim \mathcal{N}_{n,M}$, and is distributed as a Gaussian random variable $\mathcal{N}(0, c^2\sigma_T^2(h, M))$.*

*Proof.* If the column $A_{\bullet h}$ is zero, then the sketch contains no information on the signal $w$ and unbiased estimation is not possible. Otherwise, our goal is to express the unbiased sufficient statistic $T_{h,M}(A\boldsymbol{v})$ for the unknown scalar $w$.

Because row operations preserve the information in a sketch, there exists an invertible matrix $G \in \mathbb{R}^{k \times k}$ such that for $A' = GA$,

- the transformed column $h$ has value $1$ in the first row and $0$ elsewhere: $A'_{1h} = 1$ and $A'_{ih} = 0$ for $i > 1$; and

- every row $A'_{iM}$ for $i > 1$ of the submatrix restricted to the noise coordinates, is either orthogonal to the first row $A'_{1M}$ or is the zero vector $\mathbf{0}$.

Since $G$ is invertible, $A'$ is equivalent to the original sketching matrix $A$ in that we can obtain the sketch $A'\boldsymbol{v}$ from $A\boldsymbol{v}$ and vice versa.

We specify a sufficient statistics $T$ in terms of $A'\boldsymbol{v}$ (so that $T_{h,M}(A\boldsymbol{v}) \equiv T(GA\boldsymbol{v})$). Consider the distribution of the sketch $\boldsymbol{y} = A'\boldsymbol{v} = GA\boldsymbol{v}$ for $\boldsymbol{v} = w\boldsymbol{e}_h + c\boldsymbol{u}$ where $\boldsymbol{u} \sim \mathcal{N}_{n,M}$. We have $y_1 = w + c\sum_{j\in M} A'_{ij} u_j$, hence it has distribution

$$c\mathcal{N}(w, \sigma_T^2) \,, \text{ where } \sigma_T^2 := \|A'_{1M}\|_2^2 \sigma^2 = \frac{1}{m}\|A'_{1M}\|_2^2 \,. \tag{7}$$

From orthogonality of the rows $A'_{iM}$ restricted to $M$, the random variables $y_i$ for $i > 1$ are independent of $A'_{1\bullet}\boldsymbol{u}$ and hence convey no information on $w$. The information on the signal $w$ in the sketch, and the unbiased sufficient statistic is therefore $y_1 = (GA\boldsymbol{v})_1$, which has distribution (7) and is an unbiased estimator of $w$.

To establish the claim for the additive error, note that

$$\Delta_{h,M}(cA\boldsymbol{u}) = T_{h,M}(A\boldsymbol{v}) - w = (GA\boldsymbol{v})_1 - w = (cGA\boldsymbol{u})_1 \,.$$

$\square$

## C   Proof of Lower Bound on Error Lemma

This section contains a restatement and proof of Lemma 4.5.

**Lemma 4.5** (Lower Bound on Error). *Let $A \in \mathbb{R}^{k\times(m+1)}$ be a matrix with $m \geq 20k\log^2 k$. Then, for at least $0.9$ fraction of columns $h \in [m+1]$, it holds that either $A_{\bullet h}$ is all zeros or $\sigma_T^2(h, M = [m+1]\setminus\{h\}) = \Omega\left(\frac{1}{k\log k}\right)$.*

We will use the following technical claims

**Definition C.1** (Fragile Columns). Let $A \in \mathbb{R}^{k\times m}$. For each column $h \in [m]$, define

$$K_h = \{i \in [k] : A_{ih} \neq 0\}, \qquad b_{ih} = \left|\{j \in [m] : A_{ij}^2 \geq A_{ih}^2\}\right|$$

the set $K_h$ of active rows and the dominated number $b_{ih}$ of each active row. Let $i \in K_h$

$$c_1^{(h)} \leq c_2^{(h)} \leq \cdots \leq c_{|K_h|}^{(h)}$$

be the nondecreasing rearrangement of $\{b_{ih} : i \in K_h\}$). We say that a column $h$ is *fragile*, if:

$$\forall 1 \leq i \leq |K_h| : \quad c_i^{(h)} \geq \frac{i\,m}{10k\log_2 k}.$$

Note that zero columns are fragile by definition.

**Claim C.2** (Most Columns are Fragile). *If $m > 2k\log_2 k$ then at least $0.9m$ of the columns are fragile.*

*Proof.* We look at $\lfloor\log_2 k\rfloor$ ranges of $b_{ih}$ values where $R_1 = [1,\ldots,\frac{m}{10k\log_2 k})$ and $R_t = [2^{t-2}, 2^t)\frac{m}{10k\log_2 k}$ for $t \geq 2$.

We say a column $h$ is *strong* for range $t$ if it has $2^{t-1}$ or more $b_{ih}$ values in $R_t$. That is, $|\{i : b_{ih} \in R_t\}| \geq 2^{t-1}$. Clearly if a column $h$ is not fragile then it must be strong for some range $t$, but the converse may not hold.

We bound the number of distinct columns that are strong for at least one range $t$. This bounds the total number of non-fragile columns.

Consider range $t = 1$. Each of the $k$ rows contributes the columns of its $\frac{m}{10k\log_2 k}$) largest entries. So the total number of columns that can be strong for range $t = 1$ is at most $\frac{m}{10\log_2 k}$).

Now consider $t > 1$. Each row has at most $2^{t-1}\frac{m}{10k\log_2 k}$ distinct columns $h$ with $b_{ih} \in R_t$. But for a column to be strong for $t$ it has to participate in $2^{t-1}$ such rows. So in total, the range contributes at most $\frac{m}{10\log_2 k}$) strong columns.

Summing over all ranges, we obtain a bound of $m/10$ on the total number of columns that are strong for at least one range. This also bounds the number of non-fragile columns. This concludes the proof. $\qquad\square$

**Claim C.3** (Nearly to fully orthogonal rows). Let $\boldsymbol{v}^{(1)}, \ldots, \boldsymbol{v}^{(k)} \in \mathbb{R}^n$ for $k \geq 1$ be linearly independent and satisfy

$$\langle \boldsymbol{v}^{(i)}, \boldsymbol{v}^{(j)} \rangle = 1 \quad (i \neq j), \qquad \|\boldsymbol{v}^{(i)}\|^2 > i(2 + \ln k) \quad (i = 1, \ldots, k).$$

Then there are orthogonal $\boldsymbol{u}^{(1)}, \ldots, \boldsymbol{u}^{(k)}$ such that $\boldsymbol{u}^{(1)} = \boldsymbol{v}^{(1)}$, and

$$\|\boldsymbol{u}^{(i)}\|_2^2 > \|\boldsymbol{v}^{(i)}\|_2^2 - 1 \geq i(1 + \ln k) \quad (i = 2, \ldots, k)$$
$$\boldsymbol{u}^{(i)} \text{ is an affine combination of } \boldsymbol{v}^{(1)}, \ldots, \boldsymbol{v}^{(i-1)} \quad (i = 2, \ldots, k)$$

*Proof.* We construct vectors $\boldsymbol{u}^{(2)}, \ldots, \boldsymbol{u}^{(k)}$ in order using the following operations:

$$\tilde{\boldsymbol{u}}^{(i)} = \boldsymbol{v}^{(i)} - \sum_{j=1}^{i-1} \frac{\langle \boldsymbol{v}^{(i)}, \boldsymbol{u}^{(j)} \rangle}{\|\boldsymbol{u}^{(j)}\|^2} \boldsymbol{u}^{(j)}$$

$$\boldsymbol{u}^{(i)} = \frac{\tilde{\boldsymbol{u}}^{(j)}}{1 - \sum_{j=1}^{i-1} \frac{\langle \boldsymbol{v}^{(i)}, \boldsymbol{u}^{(j)} \rangle}{\|\boldsymbol{u}^{(j)}\|^2}} .$$

We establish the claim by induction on $i$. The claim clearly holds for $i = 1$.

Assume it holds for $i$. Then $\boldsymbol{u}^{(i)} = \sum_{j=1}^{i} \gamma_{ij} \boldsymbol{v}^{(j)}$, where $\sum_{j=1}^{i} \gamma_{ij} = 1$.

Therefore for all $h > i$,

$$\langle \boldsymbol{v}^{(h)}, \boldsymbol{u}^{(i)} \rangle = \sum_{j=1}^{i} \gamma_{ij} \langle \boldsymbol{v}^{(h)} \boldsymbol{v}^{(j)} \rangle = \sum_{j=1}^{i} \gamma_{ij} = 1 .$$

Therefore

$$\tilde{\boldsymbol{u}}^{(i+1)} = \boldsymbol{v}^{(i+1)} - \sum_{j=1}^{i} \frac{1}{\|\boldsymbol{u}^{(j)}\|^2} \boldsymbol{u}^{(j)}$$

Using orthogonality of $(\boldsymbol{u}^{(j)})_{j \leq i}$ we obtain:

$$\|\tilde{\boldsymbol{u}}^{(i)}\|_2^2 = \|\boldsymbol{v}^{(i)}\|_2^2 - \sum_{j=1}^{i-1} \frac{1}{\|\boldsymbol{u}^{(j)}\|_2^2} \geq \|\boldsymbol{v}^{(i)}\|_2^2 - \sum_{j=1}^{i-1} \frac{1}{i(1 + \ln k)} \geq \|\boldsymbol{v}^{(i)}\|_2^2 - 1 .$$

(using an upper bound on the Harmonic sum).

The scaling of $\tilde{\boldsymbol{u}}^{(i)}$ ensures that $\boldsymbol{u}^{(i)}$ is an affine combination of $\boldsymbol{v}^{(i)}$ and $(\boldsymbol{u}^{(j)})_{j<i}$. Since by induction each $\boldsymbol{u}^{(j)}$ for $j < i$ is an affine combination of $(\boldsymbol{v}^{(h)})_{h \leq j}$, combining we obtain that so is $\boldsymbol{u}^{(i)}$.

The scale factor is $s_i = 1 - \sum_{j=1}^{i} \frac{1}{j(1 + \ln k)} \in (0, 1]$. Therefore,

$$\|\boldsymbol{u}^{(i)}\|_2^2 = \frac{1}{s_i^2} \|\tilde{\boldsymbol{u}}^{(i)}\|_2^2 \geq \|\tilde{\boldsymbol{u}}^{(i)}\|_2^2 \geq \|\boldsymbol{v}^{(i)}\|_2^2 - 1 .$$

$\qquad\square$

*Proof of Lemma 4.5.* We first give a characterization of $\sigma_T^2(h, [m+1] \setminus \{h\})$ that we will use, and follows from a similar argument to the proof of Lemma 4.2. Let $G \in \mathbb{R}^{k \times k}$ be invertible so that the column $GA_{\bullet h}$ has only values in $\{0, 1\}$ and the rows of the submatrix $GA_{\bullet, [m+1] \setminus \{h\}}$ with column $h$ removed are orthogonal. Then

$$\frac{1}{\sigma_T^2(h, [m+1] \setminus \{h\})} = \sum_{i: GA_{ih} = 1} \frac{m}{\|GA_{i, [m+1] \setminus \{h\}}\|_2^2} . \tag{8}$$

We assume, without loss of generality, that the rows of $A$ are either orthogonal or $\mathbf{0}$. From Claim C.2, it follows that most columns of $A$ are fragile.

We now fix a nonzero fragile column $h$. We scale the rows so that $A_{\bullet h}$ are in $\{0, 1\}$. Note that the fragility of columns is invariant to rescaling of rows. From fragility, and with the rescaling, the squared norms of the rows in increasing order are at least $\frac{im}{10k \log k}$.

We now consider the submatrix $B = A_{L_h, [m+1] \setminus \{h\}} \in \mathbb{R}^{k \times m}$ of $A$ with column $h$ removed and all rows in which column $h$ was not active are removed. Let the rows of $B$ be $\boldsymbol{v}^{(i)}$ and observe that from orthogonality of the rows of $A$ and from the fragility of $h$, the vectors satisfy the conditions in the statement of Claim C.3. Therefore by applying the claim we obtain a matrix $B'$ (with row vectors $\boldsymbol{u}^i$ that are orthogonal, are affine transformations of the rows of $B$, and that

$$\sum_i \frac{1}{\|B'_{i\bullet}\|_2^2} \leq \sum_i \frac{10k \log k}{im} = O(\frac{k \log k}{m}) \ .$$

It follows that if we applied the same transformations with the column $h$, the column would be invariant. The matrix $A'$ with column $A$ with $B'$ substituting the matrix $B$ could be obtained using the same transformation. This matrix satisfies the conditions of the characterization and by applying (8) we obtain $\sigma_T^2(h, [m + 1] \setminus \{h\} = \Omega(\frac{1}{k \log k})$ and this concludes the proof.

$\square$

# D   Proof of the Gain Lemma

This section contains a restatement and proof of Lemma 4.9.

Because $T_{h,M}$ is a sufficient statistic for $w$, the distribution of the sketch $A\boldsymbol{v}$, conditioned on $T_{h,M}(A\boldsymbol{v}) = (G^{(h,M)}A\boldsymbol{v})_1 = \tau$, does not depend on $w$. Let $f_\tau : \mathbb{R}^k$ be the density function of this distribution.

We can thus express the expected value of $\psi$, conditioned on the value of the statistic $T_{h,M}(\boldsymbol{y}) = \tau$, (as it does not depend on the signal value $w$):

$$\Psi(\tau) := \int_{\mathbb{R}^k} \psi(\boldsymbol{y}) f_\tau(\boldsymbol{y}) \, d\boldsymbol{y} \ . \tag{9}$$

We express the error rate of $\psi$ (see Definition 4.8) in terms of $\Psi$ and $\sigma_T^2$:

$$\mathsf{err}(\psi) = \int_a^1 \int_{\mathbb{R}} \frac{\Psi(w + x) + 1}{2} \varphi_{0, c^2 \sigma_T^2}(x) \, dx \, \nu(w) \, dw \tag{10}$$
$$+ \int_{1+\alpha}^b \int_{\mathbb{R}} \frac{1 - \Psi(w + x)}{2} \varphi_{0, c^2 \sigma_T^2}(x) \, dx \, \nu(w) \, dw \ ,$$

where $\nu(w)$ be the density function of the distribution $W$ on the signal $w$ (see Definition 3.3) and $\phi_{0, c\sigma_T^2}$ is the distribution of the deviation (see Lemma 4.2).

**Lemma 4.9** (Gain Lemma). *If* $\mathsf{err}[\psi] < \delta$ *and attack parameters are as in Definition 3.3, and* $\sigma_T < \alpha/(c\sqrt{\log(1/\delta)})$ *then*

$$\mathsf{E}_{\boldsymbol{v}} [\psi(A\boldsymbol{v})(T_{h,M}(A\boldsymbol{v}) - w)] = \Omega(\alpha^{-1} c^3 \sigma_T(h, M)^2)$$

*Proof.* We bound from below the expected value, over our query distribution, of

$$\psi(A\boldsymbol{v})(T_{h,M}(A\boldsymbol{v}) - w) = c\psi(A\boldsymbol{v})\Delta_{h,M}(A\boldsymbol{u}).$$

We express the expected value of $\psi$ over our query distribution, conditioned on the deviation $\Delta_{h,M}(cA\boldsymbol{u}) = x$:

$$\mathsf{E}\left[\psi(A\boldsymbol{v}) \mid \Delta_{h,M}(cA\boldsymbol{u}) = x\right] = \frac{1}{\varphi_{0,c^2\sigma_T^2}(x)} \cdot \int_a^b \Psi(w+x)\varphi_{0,c^2\sigma_T^2}(x)\,\nu(w)\,dw$$

$$= \int_a^b \Psi(w+x)\,\nu(w)\,dw$$

$$= \int_{a+x}^{b+x} \Psi(w)\,\nu(w-x)\,dw \,. \tag{11}$$

We express the expected value of $\psi(A\boldsymbol{v})\Delta_{h,M}(cA\boldsymbol{u})$ over our query distribution.

$$\mathsf{E}_{\boldsymbol{v}}[\psi(A\boldsymbol{v})\Delta_{h,M}(cA\boldsymbol{u})]$$

$$= \mathsf{E}_{x\sim\mathcal{N}(0,c^2\sigma_T^2)} x \,\mathsf{E}_{\boldsymbol{v}}\left[\psi(A\boldsymbol{v}) \mid \Delta_{h,M}(cA\boldsymbol{u}) = x\right]$$

$$= \int_{-\infty}^{\infty} x\varphi_{0,c^2\sigma_T^2}(x)\left(\int_{a+x}^{b+x} \Psi(w)\,\nu(w-x)\,dw\right)dx \qquad ; \text{ using (11)}$$

$$= \int_0^{\infty} x\varphi_{0,c^2\sigma_T^2}(x)\left(\int_{a+x}^{b+x} \Psi(w)\,\nu(w-x)\,dw - \int_{a-x}^{b-x} \Psi(w)\,\nu(w+x)\,dw\right)dx \tag{12}$$

(using that $\varphi$ is a symmetric function).

We separately bound the parts of the integral in (12) due to $x \in [\alpha, \infty]$ and $x \in [0, \alpha]$.

We first consider $x \in [\alpha, \infty]$. Since $\nu$ is a density function and $|\Psi| \leq 1$, the absolute value of the difference expression is bounded by 2. Hence,

$$|(12) \text{ due to } x \in [\alpha, \infty]| \leq 2 \int_\alpha^\infty x\varphi_{0,c^2\sigma_T^2}(x)\,dx = \frac{2}{\sqrt{2\pi}}\exp(-(\alpha/(c\sigma_T))^2/2)\,.$$

We now bound the contribution to (12) due to $x \in [0, \alpha]$. We use the following:

**Claim D.1.** For $x \in [0, \alpha]$ and assuming $\mathsf{err}(\psi) \leq 0.05$ and our setting of $a, b$,

$$\int_{a+x}^{b+x} \Psi(w)\,\nu(w-x)\,dw - \int_{a-x}^{b-x} \Psi(w)\,\nu(w+x)\,dw = \Omega(c\,\alpha^{-1}\,x)\,. \tag{13}$$

*Proof.* We break the range of integration into parts according to the density function $\nu$ (see Definition 3.3) and bound each part.

$$\int_{a+x}^{b+x} \Psi(w)\,\nu(w-x)\,dw - \int_{a-x}^{b-x} \Psi(w)\,\nu(w+x)\,dw \tag{14}$$

$$= -\int_{a-x}^{a+x} \Psi(w)\,\nu(w+x)\,dw \tag{15}$$

$$+ \int_{a+x}^{1-x} \Psi(w)\,(\nu(w-x)-\nu(w+x))\,dw \tag{16}$$

$$+ \int_{1-x}^{1+x} \Psi(w)\,(\nu(w-x)-\nu(w+x))\,dw \tag{17}$$

$$+ \int_{1+x}^{1+\alpha-x} \Psi(w)\,(\nu(w-x)-\nu(w+x))\,dw \tag{18}$$

$$+ \int_{1+\alpha-x}^{1+\alpha+x} \Psi(w)\,(\nu(w-x)-\nu(w+x))\,dw \tag{19}$$

$$+ \int_{1+\alpha+x}^{b-x} \Psi(w)\,(\nu(w-x)-\nu(w+x))\,dw \tag{20}$$

$$+ \int_{b-x}^{b+x} \Psi(w)\,\nu(w-x)\,dw \tag{21}$$

We now bound each part. We use that $C = \Theta(c/\alpha)$ and $(1-a), b-(1+\alpha) = \Theta(\alpha/c)$.

$$|(15)| \le C\frac{4x^2}{1-a} = O(c^2\,\alpha^{-2}\,x^2)$$

$$(16) = \int_{a+x}^{1-x} \Psi(w)\,(-2x\frac{C}{1-a})\,dw = -2x\frac{C}{1-a}\int_{a+x}^{1-x}\Psi(w)\,dw \ge 0.1\,c\,\alpha^{-1}\,x$$

$$|(17)|, |(19)| \le 2Cx^2 = O(c\alpha^{-1}x^2)$$

$$(18) = 0$$

$$(20) = 2x\frac{C}{b-(1+\alpha)}\int_{1+\alpha+x}^{b-x}\Psi(w)\,dw \ge 0.1\,x\,c\,\alpha^{-1}$$

$$|(21)| \le C\frac{4x^2}{b-(1+\alpha)} = O(c^2\,\alpha^{-2}\,x^2)$$

Our bounds for (16) and (20) use that for $\Psi$ with error at most $\delta < 0.05$ and $x \in [0, \alpha]$, it holds (with our choices for $a, b$) that $\int_{a+x}^{1-x}\Psi(w)dw < -0.1(1-a)$, $\int_{1+\alpha+x}^{b-x}\Psi(w)dw > 0.1(b-(1+\alpha))$. $\qquad\square$

$$(12) \text{ due to } x \in [0,\alpha] = \int_0^\alpha x\varphi_{0,c^2\sigma_T^2}(x)\,\Omega(Cx)dx = \Omega(c^3\alpha^{-1}\sigma_T^2)\,. \tag{22}$$

Combining (22) and (13) we establish the claim in the statement of the Lemma: $(12) = \Omega(\alpha^{-1}\,c^3\sigma_T^2)$.

$\qquad\square$

# E    Signed sum of Gaussian vectors

**Lemma E.1** (Upper bound for the signed sum of Gaussian vectors). *Let $X_1, \ldots, X_r \overset{\text{i.i.d.}}{\sim} \mathcal{N}_m(0,1)$ in $\mathbb{R}^m$. Define*

$$M := \max_{s \in \{\pm 1\}^r} \left\|\sum_{i=1}^r s_i X_i\right\|_2.$$

*Then for every $t > 0$*

$$\Pr\left[ M \leq (\sqrt{r} + \sqrt{m} + t)\sqrt{r} \right] \geq 1 - 2e^{-t^2/2}.$$

*In particular,*

$$M \leq 2r + \sqrt{rm} \qquad \text{with probability } 1 - e^{-r/2}.$$

*Proof.* Collect the random vectors into the Gaussian matrix $X = \begin{bmatrix} X_1 & \ldots & X_r \end{bmatrix} \in \mathbb{R}^{m \times r}$. For any sign vector $s \in \{\pm 1\}^r$ we have

$$\sum_{i=1}^{r} s_i X_i = Xs, \qquad \text{hence} \qquad \left\| Xs \right\|_2 \leq \sigma_{\max}(X) \left\| s \right\|_2 = \sigma_{\max}(X)\sqrt{r},$$

where $\sigma_{\max}(X)$ is the largest singular value of $X$. Since this holds for all $s$, we get

$$M \leq \sigma_{\max}(X)\sqrt{r}. \tag{23}$$

For an $m \times r$ matrix with i.i.d. $\mathcal{N}(0, 1)$ entries, applying the standard concentration bound (see, e.g., Theorem 4.4.5 in Vershynin [38]) we obtain

$$\Pr\left[ \sigma_{\max}(X) \geq \sqrt{r} + \sqrt{m} + t \right] \leq 2e^{-t^2/2} \qquad (t > 0).$$

Combining this bound with (23) yields the first stated probability bound. For the simplified bound, we plug-in $t = \sqrt{r}$ in the above. $\qquad \square$

