# OpenReview forum: "The Cost of Compression: Tight Quadratic Black-Box Attacks on Sketches for $\ell_2$ Norm Estimation"
_NeurIPS.cc/2025/Conference — NeurIPS 2025 poster_

### Official Review · Reviewer_ctJN · 2025-06-23

**Clarity:** 2
**Significance:** 3
**Originality:** 4
**Rating:** 5
**Confidence:** 4

**Summary:**

This paper designs a **non-adaptive**  attack against linear sketches for the $\ell_2$ Norm Estimation problem. It starts with a batch of $\tilde{O}(k^2)$ non-adaptive queries. Upon receiving all query responses, an attack query is generated based on those responses, which is guaranteed to compromise the $\textit{optimal estimator}$. The $\tilde{O}(k^2)$ batch size nearly matches the $\Omega(k^2)$ upper bound achieved by JL transforms and AMS sketches, thus is nearly optimal.

Concretely, each query vector is sampled as $v_i = w_i e + u_i$, where $e_h$ is a random, fixed basis vector, $w_i$ is an independent weight drawn from a carefully designed distribution, and $u_i$ is an independent Gaussian noise supported on a random subset of $[n]$ of size $\tilde{O}(k^2)$. then the attack vector is a weighted sum of the noise vectors, where the weights are the query responses. The authors  prove the effectiveness of such an attack by showing that (1) the optimal estimator for the signal weight is a complete, sufficient statistic whose deviation is Gaussian, and (2) as long as the estimator does not err by too much, each query contributes an expected ''gain'' in aligning the weighted sum with an adversarial direction. This leads to the eventual failure of the estimator.

**Questions:**

1. There are many recent works on attacks against sketching / streaming. Has a non-adaptive “batch” attack model similar to the one in this paper been considered previously?

2. The paper suggests, as a future direction, reducing the Gaussian noise support from $O(k^2)$ to $O(k)$. Beyond satisfying aesthetic bounds, does a smaller support lead to any concrete efficiency or robustness benefits for the attack?

3. In adversarial streaming, Gribelyuk et al. (2025) extends L2 norm estimation lower bound to taks such as Lp norm, operator norm, and eigenvalues estimations. While this paper investigates a different attack model, does the result similarly generalizes to tasks beyond L2 norm estimation?

**Ethical Concerns:**

["NO or VERY MINOR ethics concerns only"]

**Final Justification:**

I didn't have major questions previously. My questions were mostly for curiosity, and the authors' answers are satisfying. So I will maintain my score.

**Limitations:**

Yes, the limitations are clearly discussed. The authors also point two potential future directions, which are
- designing attacks that compromise $\textit{every}$ query responder (beyond the optimal estimator considered in this paper), and
- reducing the support size of the Gaussian additive noise from $O(k^2)$ to $O(k)$.

**Paper Formatting Concerns:**

See **Strengths And Weaknesses**

**Quality:**

4

**Strengths And Weaknesses:**

**Strength**:
1. The non-adaptive ''batch'' attack model considered in this paper feels both intuitive and neat. And this paper studies the lower bound for the L2 norm estimation problem in this setting, which is arguably one of the most important problem. Although the bound holds only against ''optimal estimators'', one can imagine many use cases when the query responder is indeed an optimal estimator, thus I find the contribution of this paper clear.

2. While the construction of the attack queries is easy to understand, the analysis feels non-trivial and novel. I think the high-level idea of the ''gaining lemma''  may potentially apply to other problems under similar attack models.


**Weakness**:

I think the presentation of the paper can be improved.

1. Some terms are used before explaining. For example,
- ''Optimal estimator'' is only explained as ''tailored to A and D'' until page 6. It would be clearer to either move the explanation on Line 215, 216 to a much earlier place, or add a few more words on what specific ''optimal estimator'' is being considered, since this is central to understanding the contribution and applicability of the result.
- ''Signal gap'' appears in Lemma 4.1 without definition. Because ''signal'' already refers to $e_h$ earlier (and ''signal weight'' to $w$), the phrase “signal gap” is not self-explanatory and should be defined at first use.

2. The purpose of each lemma in Section 4 is not very intuitive. i.e., It's a bit hard to understand what's the role of them in the entire proof. I think it would be helpful to add a short overview of the proof strategy and how the lemmas interlock.

3. For the appendix, I only browsed through Appendix C and find
- multiple typos, e.g. "is order" on Line 544, "$2^{t-2}$" on Line 527, and unclosed brackets on Line 564, 572.
- Claim C.3. needs $\langle v^{(i)}, v^{(j)} \rangle = 1$ in the condition. Tracing down the proof of Lemma 4.5, Claim C.3 is applied on rows of a submatrix $B$ of $A$ containing orthogonal rows. I'm not sure how these rows satisfy the condition.


  While I believe the proofs in the appendix are correct (at least on the high level), I suggest the authors to carefully proofread.

---

> ### Author Rebuttal · Authors · 2025-07-28
>
> Thank you for your time and detailed, helpful comments! We will address all these issues and carefully proofread the final version.
>
> ---
>
> ### Q1: *“There are many recent works on attacks against sketching / streaming. Has a non-adaptive ‘batch’ attack model similar to the one in this paper been considered previously?”*
>
> **Response:**
> Prior works that use single-batch (non-adaptive) attacks include:
>
> - Cherapanamjeri and Nelson, NeurIPS 2020 (on JL with standard estimator)
> - Cohen et al., ICML 2022 and AAAI 2023 (on CountSketch and AMS sketch)
> - Ahmadian & Cohen, ICML 2024 (on MinHash cardinality sketches)
>
> We have added an explicit mention of these works when discussing this attack structure.
>
> ---
>
> ### Q2: *“The paper suggests, as a future direction, reducing the Gaussian noise support from $k^2$ to $k$. Beyond satisfying aesthetic bounds, does a smaller support lead to any concrete efficiency or robustness benefits for the attack?”*
>
> **Response:**
> Yes — since the attack constructs vectors over the support and stores a sum of these vectors, a smaller support directly reduces storage requirements for the attack.
>
> It is also worth noting that our empirical attacks were already effective with much smaller supports: for $k = 100$ we used $n = 1000$, and for $k = 1000$ we used $n = 20000$. We suspect that even smaller supports would suffice and plan to extend our experiments to verify this.
>
> ---
>
> ### Q3: *“In adversarial streaming, Gribelyuk et al. (2025) extends L2 norm estimation lower bounds to tasks such as Lp norm, operator norm, and eigenvalue estimation. While this paper investigates a different attack model, does the result similarly generalize to tasks beyond L2 norm estimation?”*
>
> **Response:**
> We did not consider other norms in this work, but this is an interesting direction for future research. We will add it as an open question in the conclusion section. We conjecture that the approach could extend to other settings — in particular, to $\ell_p$ norms with $p \in (1,2)$.

---

> > ### Comment · Reviewer_ctJN · 2025-08-04
> >
> > Thank you for your response and answers to my questions! I will maintain my score.

---

### Official Review · Reviewer_c6SG · 2025-06-28

**Clarity:** 3
**Significance:** 3
**Originality:** 3
**Rating:** 4
**Confidence:** 3

**Summary:**

This paper investigates the robustness of linear sketching methods for ℓ₂ norm estimation under black-box adversarial settings. The authors propose a universal, nonadaptive attack that applies to any sketching matrix and query responder, including randomized and adaptive ones. The attack constructs a carefully designed distribution over sparse signals with additive Gaussian noise, and—using only $O(k^2  log^2⁡k )$ queries—either induces a nontrivial error rate in the responses or constructs a vector that misleads the optimal estimator tailored to the attacker's query distribution. Theoretical analysis demonstrates that this query complexity is tight, and preliminary experiments confirm the trade-off between robustness and estimation accuracy.

**Questions:**

1. Is the $Ω(k^2)$ noise support size a theoretical necessity, or could it be reduced empirically without compromising effectiveness?

2. Consider removing the redundant restatement of Theorem 3.2 on Line 273 in Section 4.4, as it adds little new content.

3. While the authors claim that their result is “practically relevant,” the paper does not elaborate on the specific real-world scenarios or applications where such an attack might be impactful. A more concrete discussion of the practical implications of adversarial norm estimation in sketch-based systems would help contextualize the contribution.

**Ethical Concerns:**

["NO or VERY MINOR ethics concerns only"]

**Limitations:**

Yes

**Paper Formatting Concerns:**

I did not find any formatting problems with the paper.

**Quality:**

4

**Strengths And Weaknesses:**

Strengths:

1. Clear problem formulation: Definitions such as the (1, α)-gap problem (Definition 3.1) and adversarial noise vector (Definition 4.3) provide clean abstraction layers for analysis.

2. The paper offers a clean theoretical framework, with precise definitions and proofs.

3. The proposed attack is compatible with any sketching matrix and query responder, including randomized and adaptive estimators (Lines 125–126), demonstrating strong generality under minimal assumptions.

Weaknesses:

1. Some parts of the paper contain unnecessary redundancy; for example, Line 273 in Section 4.4 restates Theorem 3.2 in a way that duplicates prior content without adding new insight.

2. Proof presentation could be improved: Key results like Theorem 3.2 are explained informally with phrases like “the idea is to show...” (Line 279), lacking formal structure and transitions.

3. The attack's failure guarantee applies only to the optimal estimator aligned with the attack distribution and sketch matrix, and does not extend to arbitrary estimators, limiting its generality in practice.

---

> ### Author Rebuttal · Authors · 2025-07-28
>
> We thank the reviewer for their time and helpful comments! Our final version will address all the points listed under “weaknesses.”
>
> ---
>
> ### Q1: *“Is the $\Omega(k^2)$ noise support size a theoretical necessity, or could it be reduced empirically without compromising effectiveness?”*
>
> **Response:**
> Our empirical attacks were effective with a much smaller support size: for $k = 100$ we used $n = 1000$, and for $k = 1000$ we used $n = 20000$. It is likely that even smaller supports would suffice, and we plan to extend our experiments to verify this.
>
> We also conjecture that the theoretical bound on support size can be improved. The only part of our analysis that requires a quadratic bound (as opposed to linear in $k$) is one lemma concerning the signed sum of Gaussian vectors under a “worst-case” choice of signs. This condition may be unnecessarily strong for query responses, and tightening it remains an open problem.
>
> ---
>
> ### Q2: *“Consider removing the redundant restatement of Theorem 3.2 on Line 273 in Section 4.4, as it adds little new content.”*
>
> **Response:**
> Thank you for pointing this out — we will remove the redundant restatement in the final version.
>
> ---
>
> ### Q3: *“While the authors claim that their result is ‘practically relevant,’ the paper does not elaborate on the specific real-world scenarios or applications where such an attack might be impactful. A more concrete discussion of the practical implications of adversarial norm estimation in sketch-based systems would help contextualize the contribution.”*
>
> **Response:**
> When we describe our results as “practically relevant,” we are referring to the broader family of “sketch-based systems.” In their simplest form, sketches are used end-to-end — e.g., on network logs or streaming data — where an estimator is explicitly applied to approximate exact quantities. However, sketches also appear as **components** within more complex systems. For example, dimensionality reduction frequently appears as “bottleneck layers” in deep neural networks, where components learn to detect specific features and may implicitly estimate norms or proximities to other components or baselines.
>
> As we discuss in the paper, what we find striking is that the attack format — combining noise vectors with signs correlated to predictions to produce an adversarial direction — appears effective across domains, including highly non-linear image classifiers. Understanding this phenomenon in a unified way, identifying its intrinsic causes, and exploring potential mitigations is valuable. Norm estimation on linear sketches provides a clean setting for such exploration.

---

> > ### Comment · Reviewer_c6SG · 2025-08-06
> >
> > Thank you for the response. I trust these clarifications will be reflected in the final version to enhance readability. I will keep my score unchanged.

---

### Official Review · Reviewer_5Z34 · 2025-07-03

**Clarity:** 1
**Significance:** 3
**Originality:** 3
**Rating:** 4
**Confidence:** 2

**Summary:**

The paper presents an attack against linear sketches that aim to preserve the $\ell_2$ norm, showing that after a quadratic number of queries, norm estimation can be compromised.

Formally, the paper considers an attacker, a responder, and a fixed but unknown matrix $A \in \mathbb{R}^{k \times n}$, which is hidden from both parties. In each round, the attacker generates a query vector $v \in \mathbb{R}^n$ and submits it to a black-box system. The responder receives the sketch $A v$ and must decide whether $\|v\|_2 \le 1$ or $\|v\|_2 \ge 1 + \alpha$, for some fixed $\alpha > 0$.

The main result shows that there exists a family of distributions over $\mathbb{R}^n$ such that, if:
1. The attacker randomly selects a distribution from this family and samples vectors $v$ accordingly, and
2. The responder is aware of the distribution being used,

then after $\tilde{O}(\gamma^2 \alpha^{-2} k^2)$ rounds, with constant probability, one of the following occurs:
- The responder makes a constant fraction of errors in their binary decisions, or
- The attacker can construct a vector $u$ such that the optimal estimator (with respect to $A$ and the chosen distribution) returns a value outside the interval $(1 \pm \gamma)\|u\|_2$.

**Questions:**

1. The setting introduced on Page 2 involves only two parties: the adversary and the responder. However, the attack–defense framework appears to implicitly involve a third party—namely, a “system” that computes $A v$ for each query vector. This system acts as a black box accessible to both the adversary and the responder.

    This third entity is not formally introduced in the initial setup but becomes explicit on Page 5, where the paper describes each step of the attack (e.g., Line 166). Clarifying the role of the system earlier in the paper would help readers better understand the interaction model and the assumptions about information flow.


2. The experimental section includes standard estimators for the Johnson–Lindenstrauss (JL) transform.
How does the proposed attack compares to prior methods designed to target these estimators, as discussed in the introduction.

**Ethical Concerns:**

["NO or VERY MINOR ethics concerns only"]

**Final Justification:**

I trust these clarifications will be reflected in the final version to enhance readability. I will keep my score unchanged.

**Limitations:**

See weakness and questions.

**Quality:**

3

**Strengths And Weaknesses:**

1. Dimensionality reduction is widely used in data analysis and machine learning. Understanding when and how it fails contributes to our broader understanding of the stability and limitations of existing models.

2. The proposed attack is agnostic to the underlying matrix $A$, and the number of queries required matches existing bounds for answering adaptively chosen queries accurately.

3. The writing of the paper could be significantly improved.
    * Several notations are introduced before being properly defined, which hinders readability.
        For example:
        - Line 41: the symbol $\psi$ is used without a prior definition.
        - Line 100: the meaning of $\delta(\alpha)$ is unclear.
        - Line 120: the $W$ appears without explanation.

   * Additionally, the technical overview in lines 119–123 of the introduction refers to the algorithm's pseudocode, which is not presented until two sections later.
   This makes it difficult for the reader to follow the intuition behind the approach at an early stage.

   * The figures in the experimental section are difficult to read due to the text being too small.
   Axis labels, legends, and annotations should be enlarged to ensure readability.

---

> ### Author Rebuttal · Authors · 2025-07-28
>
> We thank the reviewer for their time and helpful comments\!
>
> ### W1 (improve the writing). We will address all comments, thank you\!
>
> ### Q1  **“The setting introduced on Page 2 involves only two parties: the adversary and the responder. However, the attack–defense framework appears to implicitly involve a third party—namely, a “system” that computes $Av$  for each query vector. This system acts as a black box accessible to both the adversary and the responder.”**
>
> Response:   We added “system” to the description of the interaction in page 2\. We initially attempted to streamline the description, but changed that as the reviewer found it confusing.
>
> ### Q2   **“The experimental section includes standard estimators for the Johnson–Lindenstrauss (JL) transform. How does the proposed attack compares to prior methods designed to target these estimators, as discussed in the introduction”**
>
> Response: We used simplified attacks, as explained, in the empirical section since we only attacked the standard and non-strategic robust estimators (we did not sample signal value).  Our simplified attack on the standard estimator is similar to the attack used in \[Cherapanamjeri and Nelson NeurIPS2020\]. One difference is that the CN20 attack was in terms of distance estimation: The generated queries  did not have a signal component; The responder compares the estimate of the distance to $e\_1$ to the estimate of the distance to $-e\_1$; and the result of the comparison determines the sign multiplier for forming an adversarial vector. The construction of the adversarial input in CN20 is also similar (and also similar to attacks from other domains including image classifiers): A signed sum of query vectors.

---

> > ### Comment · Reviewer_5Z34 · 2025-08-05
> >
> > Thank you for the response. I trust these clarifications will be reflected in the final version to enhance readability. I will keep my score unchanged.

---

> > > ### Author Response · Authors · 2025-08-09
> > >
> > > Thank you! The final version will include these clarifications.

---

### Official Review · Reviewer_wnnW · 2025-07-08

**Clarity:** 3
**Significance:** 3
**Originality:** 2
**Rating:** 4
**Confidence:** 1

**Summary:**

- This paper studies the problem of constructing adversarial inputs to linear sketches for the task of L2 norm estimation
- The presented attack is a universal, non-adaptive attack using $\tilde{O}(k^2)$ queries, where $k$ is the dimension of lower-dimensional projection
- Compared to prior work, this paper works for any sketching matrix (whereas prior work focused on JL/AMS constructions), and also is universal (whereas previous work may only work for some estimators)

**Questions:**

See weaknesses

**Ethical Concerns:**

["NO or VERY MINOR ethics concerns only"]

**Final Justification:**

I have read the other rebuttals/responses. The authors have clarified concerns from all reviewers and clarified the future works sections, and there seems to be a consensus that the contributions of this paper are significant.

**Limitations:**

See weaknesses

**Paper Formatting Concerns:**

No formatting concerns

**Quality:**

3

**Strengths And Weaknesses:**

Strengths
- Good presentation
- Clear contribution over prior work, reducing the number of queries for general sketching matrices to quadratic (whereas prior work work used more specific information about the sketching matrix)


Weaknesses
- Empirical verification only done for JL sketching matrices with Standard/Robust Estimator, should at least include some other sketch constructions
- Attack requires that the estimator is optimal with respect to $\mathcal{D}$ and $A$. How this can be generalized to non-optimal responders is not clear

---

> ### Author Rebuttal · Authors · 2025-07-28
>
> We thank the reviewer for their time and helpful comments!
>
> ### W1 **“Empirical verification only done for JL sketching matrices with Standard/Robust Estimator, should at least include some other sketch constructions”**
>
> We plan to add experiments with AMS sketching matrices and the standard (median based) and robustified estimators. We expect to see similar behavior to JL.
>
> ### W2 **“Attack requires that the estimator is optimal with respect to $D$ and $A$. How this can be generalized to non-optimal responders is not clear”**
>
> The attack works against *any* responder, even a strategic one with full knowledge of the attack strategy. However, the product of the attack is statistically guaranteed to only fail the optimal estimator with respect to $D$ and $A$.  A nice property of our attack is that it works in a very limited model of single batch (=  non adaptive) and produces a single adversarial input.  To be effective against strategic estimators that are tailored to the attack strategy and adapt to the actual queries, we must use fully adaptive attacks with $O(k)$ rounds of adaptivity. It is an open question whether there exists such a quadratic size attack.
>
> That said, we suspect that our current attack is effective against a wider class of non-strategic (fixed) estimators. To establish this with this form of attacks (that remarkably are similar to attacks on image classifiers) we need (for the final compromised estimator) to be able to relate its responses on queries in a weak way to the response on their linear combination. Linear estimators do that, but the hope is to specify a more general condition. These are questions for followup works.

---

> > ### Comment · Reviewer_wnnW · 2025-08-05
> >
> > Thank you for the response to the review. After reading the other reviews, I am inclined to maintain my recommendation for acceptance.

---

### Note · Authors · 2025-08-12

**Author Final Summary**

Our paper studies the robustness of **linear sketches for $\ell_2$ norm estimation** under *black-box adversarial attacks*. We present a **universal, non-adaptive, single-batch attack** that works against **any** fixed sketching matrix and **any** query responder—including randomized or adaptive ones. Using only $\tilde{O}(k^2)$ queries, the attack either forces a constant error rate or produces an adversarial vector on which the **optimal estimator** for the attack’s query distribution fails. This matches the known $\Theta(k^2)$ upper bound for JL/AMS matrices with specialized estimators, establishing *tight* quadratic bounds in a general setting.

**Key contributions and novelty:**
1. **General applicability** — previous tight attacks focused on specific sketches/estimators; ours works for any linear sketch and any responder.
2. **Clean theoretical framework** — the $(1,\alpha)$-gap abstraction and “gain lemma” isolate the structure enabling the attack.
3. **Structural parallels to other domains** — our attack’s format closely resembles adversarial examples in image classification, suggesting a unifying view of vulnerabilities in compressed representations.

**Clarifications and updates (per reviews):**
- We will broaden experiments to **AMS** sketches (in addition to JL) with both standard and robust estimators, where we expect similar trends. We will also explore the needed support size.
- We will improve exposition: introduce all parties (including the “system”) early, define key terms at first use (“optimal estimator,” “signal gap”), streamline proof presentation, and enlarge figure text for readability.
- We will remove redundancy (e.g., repeated theorem restatements) and proofread the appendix for typos and missing conditions.

**Future directions:**
- Reducing Gaussian noise support (empirically already much smaller than theoretical bound) for efficiency gains.
- Design attacks of quadratic size that compromise any strategic estimator (requires multiple batches; our result is the best possible in a single batch)
- Extending techniques to other norms ($\ell_p$, $p\in(1,2)$), operator norms, and related sketch-based tasks.
- Understanding why similar adversarial constructions succeed across linear sketches and nonlinear classifiers, and developing principled mitigations.

We thank the reviewers and ACs for constructive feedback and will incorporate all suggested clarifications and improvements into the final version.

---

### Decision · Program_Chairs · 2025-09-17

**Decision:**

Accept (poster)

**Comment:**

This paper studies the robustness of linear sketches for ℓ₂ norm estimation under black-box adversarial attacks. The authors present a universal, non-adaptive, quadratic-size attack that succeeds against *any* linear sketching matrix and query responder, including randomized or adaptive ones. The result matches known upper bounds achieved for specific sketches (e.g., JL, AMS) with specialized estimators. The paper develops a clean theoretical framework, introducing abstractions such as the gap problem and the “gain lemma,” and provides preliminary experiments.

**Strengths:**
- Establishes a tight quadratic lower bound for adversarial attacks in a general sketching setting, extending beyond prior results.
- Generality: the attack applies to any linear sketch and responder, independent of distributional assumptions.
- The theoretical development is constructive, with well-isolated abstractions that clarify the attack mechanism.
- Empirical results, though limited, illustrate the attack in practice and confirm the expected failure modes.

**Weaknesses:**
- Empirical validation is somewhat limited (mainly to JL sketches with standard/robust estimators); experiments with other sketches (e.g., AMS) would strengthen the work.
- Exposition could be improved: some definitions (e.g., “system,” “optimal estimator,” “signal gap”) appear late or without sufficient clarity; figures are hard to read; the appendix contains redundancy and typos.
- The main failure guarantee applies to the optimal estimator under the attack distribution; extending guarantees to arbitrary or strategic estimators remains an open problem.

**Outcomes of rebuttal:**
The author–reviewer exchanges were constructive/productive. Authors committed to clarifying the exposition, introducing all notions or players (including the “system”), and improving figures and proof readability. They also promised to expand experiments to AMS sketches and explore smaller noise support sizes. Reviewers generally found the theoretical contribution solid and significant, and maintained or slightly increased their scores after rebuttal. One reviewer signaled lack of expertise, but others provided substantial evaluations.

**Summary:**
This paper makes a clear theoretical contribution to the adversarial robustness of low-dimensional representations, proving tight quadratic lower bounds for adversarial attacks on sketch-based norm estimation. While the exposition and empirical scope can be improved, the contribution is significant. I recommend acceptance as a poster, and encourage the authors to incorporate the clarifications and additional experiments discussed during the rebuttal.